# Benchmarking State-of-the-Art Approaches for Norovirus Genome Assembly in Metagenome Sample

**DOI:** 10.3390/biology12081066

**Published:** 2023-07-29

**Authors:** Dmitry Meleshko, Anton Korobeynikov

**Affiliations:** 1Center for Algorithmic Biotechnology, St. Petersburg State University, 7/9 Universitetskaya Emb., 199004 St. Petersburg, Russia; 2Department of Statistical Modelling, St. Petersburg State University, Universitetskiy 28, 198504 St. Petersburg, Russia

**Keywords:** benchmarking, viral assembly, noroviruses

## Abstract

**Simple Summary:**

Proper use of state-of-the-art tools allows for complete recovery of viral genomes from a metagenome sample.

**Abstract:**

A recently published article in BMCGenomics by Fuentes-Trillo et al. contains a comparison of assembly approaches of several noroviral samples via different tools and preprocessing strategies. It turned out that the study used outdated versions of tools as well as tools that were not designed for the viral assembly task. In order to improve the suboptimal assemblies, authors suggested different sophisticated preprocessing strategies that seem to make only minor contributions to the results. We have reproduced the analysis using state-of-the-art tools designed for viral assembly, and we demonstrate that tools from the SPAdes toolkit (rnaviralSPAdes and coronaSPAdes) allow one to assemble the samples from the original study into a single contig without any additional preprocessing.

## 1. Introduction

In recent years, there has been a significant surge of interest in the field of viral discovery, with numerous large-scale studies being conducted [1,2,3]. Many research groups have shifted their focus to viral studies, therefore exploding the amount of papers devoted to the subject. Despite the considerable attention given to viral genome assembly, there is currently no established state-of-the-art method for this task. The diversity of viral genomes and corresponding sequencing data presents a challenge, as assembly approaches suitable for dsDNA bacteriophages cannot be directly applied to RNA viruses or retroviruses without appropriate modifications. As a result, some papers are naturally devoted to benchmarking viral assembly approaches for different kinds of input data. However, proper benchmarking design (cf. [4,5,6,7] as an example of well-designed studies for non-viral data) and interpretation of the obtained results is a non-trivial task, and subtle flaws made here could easily lead to misleading or controversial outcomes.

An article discussing different approaches for norovirus genome assembly was recently published in BMC Genomics [1]. Using the approaches presented in this article, we want to highlight some common benchmarking problems that might result in misleading conclusions. By highlighting these problems, we hope to emphasize the importance of avoiding such pitfalls in future studies.

The article was submitted and subsequently published in 2021; however, the tools utilized in the study were notably outdated. Specifically, a comparison of genome assemblers metaSPAdes v.3.11.1 [8] and MEGAHIT v.1.1.3 [9] was conducted. Notably, metaSPAdes 3.11.1 was released in March 2018, while the current version of SPAdes is 3.15.5, and the SPAdes team makes multiple releases each year. SPAdes 3.15.0 was released on January 2021 and was available to use in this study. Similarly, MEGAHIT v.1.1.3 was released in March 2018, while the most recent version, 1.2.9, was released in October 2019. In scientific research, the use of legacy tool versions can be justified based on reasons such as reproducibility, reliability, and compatibility between different tools and pipelines. Moreover, the extended duration of the scientific publishing process can make it impractical to update the tool versions used at every revision, which can take months in some cases. However, in the case of this article, the utilization of excessively outdated tool versions raises concerns because of the substantial time gap between the release of the tool versions employed and the time of publishing. This time gap undermines the applicability of any of the aforementioned reasons for legacy software usage as a valid justification. Therefore, it is important to critically evaluate the impact of using outdated tools alongside claims of superior performance, as this approach may lead to potential misinterpretations and fail to accurately represent the current state of the art. This concern is particularly pertinent in benchmarking studies, where the comparison of tools constitutes the primary outcome of the research, compared to papers that offer novel biological insights.

Moreover, the selection of tools for assembly benchmarking raises questions, and no further justification is provided for this choice. It is worth noting that both metaSPAdes and MEGAHIT are metagenomic assemblers, and while they have shown their effectiveness in non-metagenomic settings, including viral assemblies [10,11], the absence from the study of specialized assemblers designed for RNA and RNA viral data is concerning and may indicate an improper benchmark design. Given that noroviruses are RNA viruses, it is surprising that no dedicated transcriptome or viral assembler was evaluated. Long before the time of the study, there were several prominent assemblers readily available for consideration, such as Trinity  [12], rnaSPAdes  [13] (included in the used SPAdes v.3.11.1 package), Savage [14], and IVA [15], among others. Moreover, a dedicated RNA viral assembler, rnaviralSPAdes  [16], had been developed, with the preprint and the tool itself being publicly available since the summer of 2020.

Furthermore, it is worth noting that the authors of the study did not adhere to the recommended best practices for genome assembly as outlined in the SPAdes and MEGAHIT manuals. One of the important guidelines is to select a *k*-mer length that is approximately half the length of the input sequencing reads. However, the authors reported a mean read length of 100 bp and tested various *k*-mer lengths, including extremely small values such as 21 and exceptionally long values such as 99 and 119. It is important to highlight that reporting these exceedingly long *k*-mer values can create a false impression that tools such as MEGAHIT and metaSPAdes are unable to handle such *k*-mer lengths effectively. In reality, these values are not suited for the given read length, since it is impossible to build a meaningful de Bruijn graph with such parameters from these input data.

Finally, the presentation of several statistics that the authors used to show the results could be improved. The study involved the assembly of a total of eight datasets, and the authors reported mean assembly statistics values across these datasets. From experience, these values might vary a lot between assembly runs. Relying solely on mean N50, contig size, and the number of contigs provides limited information about the overall assembly results. Furthermore, different assemblers may employ distinct heuristics for contig filtering, making the direct comparison of these values for benchmarked approaches challenging, especially in a metagenomic context.

To provide a more comprehensive assessment, it would be beneficial to reformat the resulting tables, allowing readers to compare various statistics across multiple datasets and evaluate their variability. While the authors partially addressed this issue by aligning contigs using BLAST and reporting mean values, it is crucial to consider the limited sample size of only eight datasets. By presenting the statistics in a format that facilitates comparison across datasets, readers can gain a better understanding of the performance variation and make more informed assessments of the benchmarked approaches.

To showcase the effect of tool choice, we reproduced the analysis from the mentioned study using the state-of-the-art version of tools and showed that the majority of them do correctly assemble the noroviral datasets in question into a single contig without requiring additional complex clustering procedures that were previously suggested in the article as a means to address assembly result deficiencies. It is worth noting that the benchmarks performed by the original authors are readily reproducible, as the necessary data and the versions of the tools used were explicitly stated.

## 2. Methods

The raw data obtained in the original study [1] consist of eight fecal samples collected from patients affected by acute non-bacterial gastroenteritis. The samples were subjected to polyA enrichment using TruSeq RNA Sample Prep Kit v2, followed by Illumina HiSeq 2000 sequencing. For each sample, 15–69 million paired reads of 100 bp were generated. Input data were downloaded from NCBI SRA.

For viral assembly, benchmarking input reads were trimmed using BBDuk with trimpolya=15 qtrim=rl trimq=10 parameters, as in [16], and assembled using coronaSPAdes 3.15.4, rnaviralSPAdes 3.15.4, rnaSPAdes 3.15.4, Trinity 2.13.2, and MEGAHIT 1.2.9, with the default parameters including k-mer sizes. Exact command lines are available in the Appendix A. The assembly was performed on a local server with Intel Xeon E7-4800 v2 CPU using 16 computational threads.

coronaSPAdes requires a curated set of HMM profiles that matches a particular viral family. While the sets of coronaviral, HIV, and influenza HMMs are readily available from the coronaSPAdes website, most of the viral families (including noroviruses) were left uncovered.

Since noroviruses are small (≈ 7500 nt) and have quite simple internal genome organization comprising 3 almost non-overlapping ORFs that encode several polyproteins [17], we prepared a set of norovirus HMM models for coronaSPAdes, extracting the models from the RVDB-prot-HMM database [18] with “norovirus” in the keywords. In total, 13 HMM models were obtained, representing different protein families of norovirus genomes for sequences deposited in GenBank.

Scaffolds produced by each assembler were aligned to corresponding references using QUAST v 5.0.2 [19].

## 3. Results

The quality assessment results obtained are summarized in Table 1. They clearly show that rnaviralSPAdes and coronaSPAdes (the version of rnaviralSPAdes that could use profile HMM models to guide an assembly) are better suited for the assembly of these data than rnaSPAdes, as well as metaSPAdes and MEGAHIT (the last two were benchmarked by [1]). We note that each sample was assembled into a single contig with perfect or near-perfect quality.

Moreover, the results clearly show that one does not need any additional pre- and post-processing steps beyond quality trimming to obtain decent results, contrasting with the complicated assembly pipeline presented in [1] that included multiple steps of read binning, contamination filtering, and norovirus read filtering. All these steps might influence the final result and cause the degradation of assembly quality in general.

Since our approach only includes read trimming as a preprocessing step, and therefore can be directly compared to *pC* approach of [1], here we see that coronaSPAdes was able to assemble seven out of eight samples into a contig longer than 7500 nt, and the contig length of the remaining sample is 7493 nt, which places this sample into a near-complete category. The original *pC* approach utilized metaSPAdes and MEGAHIT, which assembled four out of eight and five out of eight samples into contigs longer than 7500 nt, respectively. We summarized these results in Figure 1.

Further analysis might include checking the assembly graph to figure out if these assembly length differences are due to the assembler being over-aggressive in eliminating sequencing errors, or due, e.g., to uncollapsed viral quasi-species variation [16].

## 4. Discussion

The genome assembly task is a very challenging but quite well-studied computational problem. Multiple genome assemblers are available, and the choice of the particular tool is highly dependent on the input data properties and even the result desired. Nevertheless, even the proper choice of an assembler suitable for a given kind of input data cannot guarantee a complete genome assembly for complex datasets. However, we emphasize that even for datasets with low complexity, the researcher should choose an assembler carefully. A common problem that is seen in many papers is the usage of improper tools, improper options, or obsolete versions of the tools.

We showed that specialized RNA viral assemblers such as coronaSPAdes and rnaviralSPAdes were able to outperform metagenomic assemblers metaSPAdes and MEGAHIT and transcriptomic assemblers Trinity and rnaSPAdes in the noroviral assembly task. While the two latter assemblers were designed to cope with the challenges of RNA sequencing data [13], their aim is completely different as they are trying to extract all possible valid transcript; therefore, their results might be suboptimal in the presence of multiple viral strains, quasispecies, and splice events [20].

We specifically outline that while coronaSPAdes is able to improve viral assemblies, it requires a curated set of HMM profiles that matches a particular viral family. In addition to noroviral HMMs, we have already prepared coronaviral, HIV, and influenza HMMs (which are available for download from the coronaSPAdes website), but most of the viral families were left uncovered. However, the procedure for creating a set of HMM profiles for a particular viral family is not a particularly complicated process.

Here, we provide some guidance on creating user-defined sets of HMMs. We recommend that one obtain a set of HMMs for the viral family of interest by extracting top-matching HMMs from a reliable source such as RVDB-prot-HMM [18]. However, due to the generic naming of the HMMs in the database, the hmmfetch from the HMMER3 suite [21] may not be the most straightforward option to use. Still, in many cases, it might be possible to query the provided annotation and create a subset of viral HMMs that might be used to supplement an assembly.

It might happen that the subset of HMMs extracted by annotation from RVDB-prot-HMM is too broad and can therefore match many extraneous sequences. In such a case, it would make sense to reduce the size of the set.

One way of doing this is to align HMMs from U-RVDB to one or several reference genomes from the viral family of interest using hmmscan. The top matching HMMs that have a significant match to the reference genome are usually relevant to the specific viral family. These HMMs can be extracted from U-RVDB using hmmfetch and used in coronaSPAdes directly. Another option is to generate a custom set of HMMs if several reference genomes are available. The user can start by predicting coding sequences using tools such as Prodigal [22], GeneMark [23], or vGas [24], which are specifically designed for gene prediction and the extraction of coding sequences. Once the coding sequences have been extracted, they can be clustered using tools such as CD-HIT [25], UCLUST, MUSCLE [26], or MMSeqs [27], and HMMs can be built for each cluster using HMMER. This approach can be especially useful for studying highly diverse viral families, where standard HMMs may not be sufficient to accurately capture the diversity of the family.

Additionally, it is crucial to ensure that the resulting HMM set uniformly covers the reference genome, leaving no large gaps or uncovered segments. To achieve this, one can align the HMM set to the reference genome using hmmalign and check the alignment coordinates. It is worth noting that coronaSPAdes can handle overlapping HMMs, although the obtained results are better interpreted when different HMMs do not overlap. A customized set of HMMs is a powerful tool that researchers can use to achieve more accurate and comprehensive viral genome assemblies, particularly when working with a large number of diverse datasets.

## 5. Conclusions

Our experiments showed that genome assembly using specialized tools and the latest versions of these tools yields better results in terms of both correctness and contiguity. Ad hoc assembly approaches end in worse results. Moreover, the labor costs associated with such approaches are much higher because suboptimal results force researchers to find a way to improve the initial results, which is usually harder than the assembly itself. Finally, this short note emphasizes the importance of specialized tool development and promotion.

## Figures and Tables

**Figure 1 biology-12-01066-f001:**
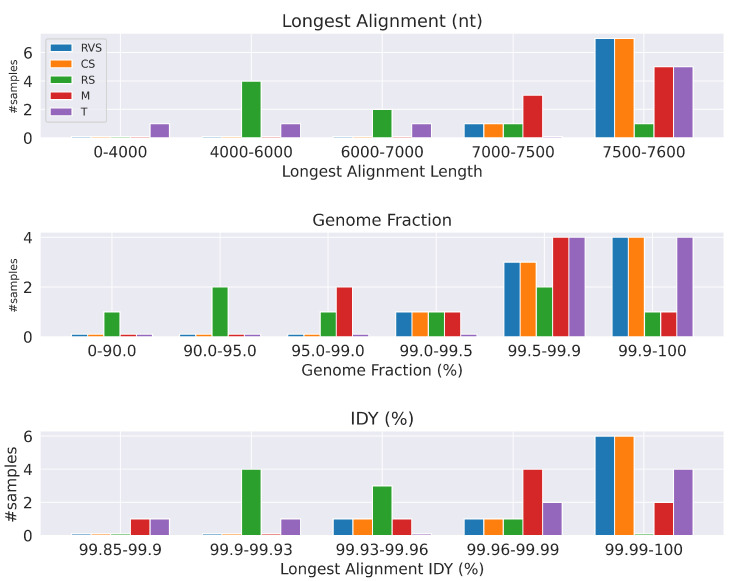
Summary of benchmarks for noroviral datasets. Tested tools are rnaviralSPAdes (RVS), coronaSPAdes (CS), rnaSPAdes (RS), MEGAHIT (M), and Trinity (T). coronaSPAdes and rnaviralSPAdes have the best performance across all metrics.

**Table 1 biology-12-01066-t001:** Benchmarking of assemblers rnaviralSPAdes (RVS), coronaSPAdes (CS), rnaSPAdes (RS), MEGAHIT (M), and Trinity (T) on several noroviral datasets. The best results are outlined in bold.

		RVS	CS	RS	M	T
SRR8074276	Longest alignment (nt)	7538	7538	7538	**7548**	**7547**
	Genome fraction%	99.83	99.83	99.83	**99.96**	**99.94**
	Longest alignment IDY%	**99.95**	**99.95**	99.91	99.87	99.93
SRR9141472	Longest alignment (nt)	**7569**	**7569**	5282	7560	5848
	Genome fraction%	**100.0**	**100.0**	78.80	99.88	**100.0**
	Longest alignment IDY%	**100.0**	**100.0**	99.95	99.99	**100.0**
SRR9141473	Longest alignment (nt)	**7536**	**7536**	4,979	7487	6,899
	Genome fraction%	**100.0**	**100.0**	90.55	99.35	**100.0**
	Longest alignment IDY%	**100.0**	**100.0**	99.91	99.97	**100.0**
SRR9141474	Longest alignment (nt)	**7541**	**7541**	6,838	7516	**7542**
	Genome fraction%	**99.99**	**99.99**	99.99	99.65	**100.0**
	Longest alignment IDY%	**100.0**	**100.0**	99.91	99.99	**100.0**
SRR9141475	Longest alignment (nt)	7482	7493	7049	7440	**7534**
	Genome fraction%	99.28	99.43	99,08	98.72	**99.97**
	Longest alignment IDY%	**100.0**	**100.0**	99.96	**100.0**	**100.0**
SRR9141476	Longest alignment (nt)	**7533**	**7533**	5699	7526	**7533**
	Genome fraction%	**100.0**	**100.0**	**100.0**	99.90	**100.0**
	Longest alignment IDY%	**100.0**	**100.0**	99.98	**100.0**	99.98
SRR9141477	Longest alignment (nt)	**7554**	**7554**	5935	7467	3658
	Genome fraction%	**99.95**	**99.95**	91.20	98.79	99.84
	Longest alignment IDY%	**100.0**	**100.0**	99.93	99.96	99.97
SRR9141478	Longest alignment (nt)	**7540**	**7540**	6,592	**7540**	**7540**
	Genome fraction%	**100.0**	**100.0**	95.90	**100.0**	**100.0**
	Longest alignment IDY%	**99.99**	**99.99**	99.95	**99.99**	99.88

## Data Availability

Noroviral HMMs that were used for coronaSPAdes assembly are available at https://cab.spbu.ru/software/coronaspades/ (accessed on 26 July 2023).

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
