# Peer review of "Benchmarking State-of-the-Art Approaches for Norovirus Genome Assembly in Metagenome Sample"

_biology, 2023, doi:10.3390/biology12081066_

Round 1

Reviewer 1 Report

In their manuscript, the authors have made efforts to address certain issues present in the paper by Fuentes-Trillo et al. I appreciate their intention to improve and replicate the results. However, it is important to acknowledge that the authors themselves have developed one of the tools discussed in the paper, which creates a potential conflict of interest. I believe it is essential for the authors to explicitly disclose this information to ensure transparency.

Regarding the criticism raised about the use of outdated tools, I suggest rephrasing it in a more nuanced manner. There can be valid reasons for utilizing legacy versions of software, and it is worth considering that the time required to finalize a manuscript, not to mention the duration of the review process, can inadvertently lead to the perception that the software used was outdated. It is important to consider these factors before making definitive statements regarding the software's relevance to the research.

Despite these concerns, I find the manuscript to be interesting and valuable to readers. To further enhance its appeal, I would strongly encourage the authors to include a graphical summary showcasing the performance of the tools discussed. This addition would not only make the manuscript more engaging, but it would also provide a visual representation of the tools' capabilities, facilitating a better understanding for readers.

Overall, while acknowledging the need for improvement, I appreciate the contributions made by the authors and believe their work holds significance in the field. By addressing the conflict of interest, refining the critique on outdated tools, and incorporating a graphical summary, the authors can enhance the quality and impact of their manuscript.

Author Response

In their manuscript, the authors have made efforts to address certain issues present in the paper by Fuentes-Trillo et al. I appreciate their intention to improve and replicate the results. However, it is important to acknowledge that the authors themselves have developed one of the tools discussed in the paper, which creates a potential conflict of interest. I believe it is essential for the authors to explicitly disclose this information to ensure transparency.

Answer: Thanks, we updated “Statement of conflicts” that we are the authors of coronaSPAdes, rnaviralSPAdes and metaSPAdes. 

TextEdit: DM and AK are authors of \coronaSPAdes,\rnaviralSPAdes, and \metaSPAdes. 

Regarding the criticism raised about the use of outdated tools, I suggest rephrasing it in a more nuanced manner. There can be valid reasons for utilizing legacy versions of software, and it is worth considering that the time required to finalize a manuscript, not to mention the duration of the review process, can inadvertently lead to the perception that the software used was outdated. It is important to consider these factors before making definitive statements regarding the software's relevance to the research.

Answer: Thanks, this is a valid point, so we extended the corresponding paragraph to discuss in more detail.

TextEdit: The article was submitted and subsequently published in 2021; however, the tools utilized in the study were notably outdated. Specifically, the comparison of genome assemblers \metaSPAdes v.3.11.1~\cite{metaspades} and \Megahit v.1.1.3~\cite{megahit} was conducted. Notably, \metaSPAdes 3.11.1 was released in March 2018, while the current version of SPAdes is 3.15.5 and the SPAdes team makes multiple releases each year. Similarly, \Megahit v.1.1.3 was released in March 2018, while the most recent version, 1.2.9, was released in October 2019. In scientific research, the use of legacy tool versions can be justified based on reasons such as reproducibility, reliability, and compatibility between different tools. Moreover, the extended duration of the scientific publishing process can make it impractical to update the tool versions used at every step. However, in the case of this particular article, the utilization of excessively outdated tool versions raises concerns regarding the validity of these reasons. The substantial time gap between the release of the tool versions employed and the time of publishing undermines the applicability of utilizing legacy versions as a valid justification. Therefore, it is important to critically evaluate the impact of using outdated tools alongside claims of superior performance, as this approach may lead to potential misinterpretations and fail to accurately represent the current state of the art. This concern is particularly pertinent in benchmarking studies, where the comparison of tools constitutes the primary outcome of the research, compared to papers that offer novel biological insights.

Despite these concerns, I find the manuscript to be interesting and valuable to readers. To further enhance its appeal, I would strongly encourage the authors to include a graphical summary showcasing the performance of the tools discussed. This addition would not only make the manuscript more engaging, but it would also provide a visual representation of the tools' capabilities, facilitating a better understanding for readers.

Answer: Thanks, we added a figure about tools performance.

TextEdit: see Figure 1

Overall, while acknowledging the need for improvement, I appreciate the contributions made by the authors and believe their work holds significance in the field. By addressing the conflict of interest, refining the critique on outdated tools, and incorporating a graphical summary, the authors can enhance the quality and impact of their manuscript.

Answer: We want to thank the reviewer again for the fair review.

Reviewer 2 Report

The manuscript entitled 'Benchmarking state-of-the-art approaches for nor-virus genome assembly in metagenome sample' by Meleshko and Korobeynikov is a brief report on a topic of general interest in the field of metagenomic virus discovery. Though I find the topic of this manuscript very interesting, the writing and structure of the manuscript detracts significantly from the work. The language throughout the manuscript is very conversational and not appropriate for scientific writing as it leaves a lot of interpretation to the reader. I would recommend restructuring of the manuscript for charity and the presentation of the results in a graphical format. 

The methods are not descriptive enough and the analysis should be repeated  as aligners do not always produce consistent results across runs. Though this is a repetition of a previous study, please provide details of the datasets used. 

Overall I think the idea here is very valid and interesting to those in the field and thus I would like to see the method improved and the results presented in a clearer manner. I think it would be a disservice to the idea behind this work to publish it as is stands. 

The paper is written in conversational English which is not appropriate for scientific publication as it leaves too much interpretation to the reader. I would recommend a complete restructuring of the manuscript and use of more precise language throughout. 

Author Response

The manuscript entitled 'Benchmarking state-of-the-art approaches for nor-virus genome assembly in metagenome sample' by Meleshko and Korobeynikov is a brief report on a topic of general interest in the field of metagenomic virus discovery. Though I find the topic of this manuscript very interesting, the writing and structure of the manuscript detracts significantly from the work. The language throughout the manuscript is very conversational and not appropriate for scientific writing as it leaves a lot of interpretation to the reader. I would recommend restructuring of the manuscript for charity and the presentation of the results in a graphical format.

Answer: Thanks for that comment, we added a graphical representations of the results. We also agree that our writing style was overly relaxed and revised the text throughout the manuscript to follow scientific writing principles.

The methods are not descriptive enough and the analysis should be repeated  as aligners do not always produce consistent results across runs. 

Answer: with all due respect, we kindly disagree with this statement. We actually don’t use read aligners at any step of our approach. QUAST internally uses minimap2 to align contigs, but QUAST is a well-established standard-de-factor tool in the area used by thousands of papers. Also, minimap2 is designed in the way to have the same results on the same input data. If you have in mind genome assemblers and not aligners, we would need to note that all SPAdes-family assemblers are designed to produce consistent results between runs. While we can’t guarantee that for MEGAHIT and TRINITY, we don’t think that rerunning the analysis will add any value to the paper. 

Though this is a repetition of a previous study, please provide details of the datasets used.

Answer: Thanks, we added a brief information about the datasets used in the paper.

TextEdit: The raw data obtained from the \cite{noropaper} study consists of eight samples collected from patients affected by acute non-bacterial gastroenteritis. The samples were subjected to polyA enrichment, followed by Illumina HiSeq sequencing. For each sample, 15-69 million paired reads of 100 bp were generated.

Overall I think the idea here is very valid and interesting to those in the field and thus I would like to see the method improved and the results presented in a clearer manner. I think it would be a disservice to the idea behind this work to publish it as is stands.

Answer: Thanks for your review, it was very helpful for improving the manuscript.